# Smart Inhibition Action of Amino Acid-Modified Layered Double Hydroxide and Its Application on Carbon Steel

**DOI:** 10.3390/molecules28155863

**Published:** 2023-08-03

**Authors:** Elena Messina, Marianna Pascucci, Cristina Riccucci, Francesca Boccaccini, Maria Teresa Blanco-Valera, Ines Garcia-Lodeiro, Gabriel Maria Ingo, Gabriella Di Carlo

**Affiliations:** 1Institute for the Study of Nanostructured Materials (ISMN), National Research Council (CNR), SP35d 9, 00010 Montelibretti, Italy; marianna.pascucci@cnr.it (M.P.); cristina.riccucci@cnr.it (C.R.); francesca.boccaccini@uniroma1.it (F.B.); gabrielmaria.ingo@cnr.it (G.M.I.); 2Department of Earth Sciences, Sapienza University of Rome, Piazzale Aldo Moro 5, 00185 Rome, Italy; 3Eduardo Torrroja Institute for Construction Science—IETCC–CSIC, Serrano Galvache 4, 28033 Madrid, Spain; blancomt@ietcc.csic.es (M.T.B.-V.); iglodeiro@ietcc.csic.es (I.G.-L.)

**Keywords:** active corrosion protection, smart nanocarrier, migrating corrosion inhibitor, green inhibitor, mortar

## Abstract

Surface impregnation of concrete structures with a migrating corrosion inhibitor is a promising and non-invasive technique for increasing the lifetime of existing structures that already show signs of corrosion attack. The main requirement for inhibitors is their ability to diffuse the rebar at a sufficient rate to protect steel. The use of smart nanocontainers such as layered double hydroxides (LDH) to store corrosion inhibitors significantly increases efficiency by providing an active protection from chloride-induced corrosion. The addition of LDH to reinforced mortar can also improve the compactness and mechanical properties of this matrix. Here, we report the synthesis of a magnesium–aluminum LDH storing glutamine amino acid as a green inhibitor (labeled as Mg–Al–Gln), which can be used as a migrating inhibitor on mortar specimens. The corrosion behavior of the specimens was determined via electrochemical techniques based on measurements of corrosion potential and electrochemical impedance spectroscopy. A cell containing a 3.5% NaCl solution was applied to the mortar surface to promote the corrosion of embedded rebars. The specimens treated with Mg–Al–Gln presented an improved corrosion protection performance, exhibiting an increase in polarization resistance (Rp) compared to the reference specimens without an inhibitor (NO INH). This effect is a consequence of a double mechanism of protection/stimuli-responsive release of glutamine and the removal of corrosive chloride species from the medium.

## 1. Introduction

In the last few decades, research efforts have furthered the study of new compounds able to prevent or hinder corrosion and our understanding of the inhibition mechanism [1,2,3]. In recent years, researchers have paid considerable attention to the development of more effective and “green” species for the corrosion protection of steel rebars. Non-commercial inhibitors have been studied, including both inorganic (phosphate [4] and molybdates [5]) and organic (benzoate [6], quaternary ammonium salts [7], and amino acid substances [8,9,10,11]) compounds.

The use of coating, which is easily applied, is one of the most common methods for the corrosion protection of metallic substrates. Corrosion inhibitors are usually added to the coating formulation to improve the protective properties. The development of protective coatings for metal substrates is of great significance for a wide range of applications, including the conservation of cultural heritage [12,13,14]. 

In recent years, some inhibitor-loaded smart carriers have been developed to protect metals from corrosion, such as nanocomposites [15], hollow mesoporous nanospheres [16], silica nanocontainers [17], and layered double hydroxide nanocontainers with organic or inorganic inhibitions [18]. Furthermore, for carbon steel rebars, one of the strategies to improve corrosion resistance is to use an inorganic protective coating dropped on the metal surface [19]. This procedure is particularly suitable for freshly prepared concrete/mortar samples, but the same cannot be said for aged samples.

Migrating corrosion inhibitors (MCIs) are products that are designed to diffuse through concrete and slow the corrosion of embedded reinforcements by forming a protective layer around them [20,21]. Because of their ability to be applied easily onto the surface and the claim that they can prevent or inhibit corrosion without change to appearance or texture, MCIs also represent an appealing option for anticorrosion treatment in concrete heritage [22].

Many researchers have added inhibitor-intercalated LDH nanocarriers to organic coatings or concrete matrices with the aim of improving the corrosion resistance of the metals. The inhibitors can migrate toward the embedded reinforcements and form a monolayer film on the surface of the steel, such as the passive layer that naturally forms on steel in an alkaline environment. This monolayer film protects the steel from corrosive processes [23]. Many substances have been tested as MCIs against the corrosion of reinforced steel bars [24]. Alkanolamine-based corrosion inhibitors were tested in addition to a common mortar and two repair mortars because they can move via diffusion through the pore structure of concrete to reach the surfaces of steel bars. The results showed that the simultaneous use of the alkanolamine-based inhibitor with a good barrier coating offers protection against rebar corrosion and allows for the restoration of damaged concrete structures [25]. It is known that organic corrosion inhibitors act by adsorption on metal surfaces, forming an organic layer that may inhibit both anodic and cathodic processes. As reported in the literature, the inhibition effectiveness of using 80 organic substances in preventing the chloride-induced corrosion of carbon steel rebars was studied by means of electrochemical potentiodynamic polarization tests in a synthetic alkaline pore solution, and amino acids showed some inhibitory effects [26,27]. The corrosion inhibition efficiency of the amino acids depends on their chemical structure. According to the literature, the best inhibition efficiency was also obtained for Gln at a 0.1 M concentration level [15]. 

Smart nanocarrier systems, i.e., LDH, were loaded with a green inhibitor, and their stimuli responsive properties were exploited to achieve active corrosion protection instead of simple passive protection [16,17]. The release mechanism from LDH species is based on anion exchange. Inhibitors intercalated in their anionic form can be replaced by chloride ions coming from the environment, and this ensures an on-demand protective action [18]. Thus, LDHs can be used with a double action: (i) the release of corrosion inhibitors; and (ii) the entrapment of aggressive species such as chlorides, ensuring the protection of metallic substrates [12,13,14,15,16,17]. Tedim et al. [28] demonstrated the possibility of intercalating Zn–Al–LDHs with three inhibitors, stressing the versatility of nanocontainer structures in incorporating various organic and inorganic anions. The protective effect provided by mixed LDHs was investigated on bare AA2024 substrates immersed in electrolyte solutions. Nitrite released from the same Zn–Al–LDH was found to be sensitive to chloride concentration in the environment, resulting in a higher long-term inhibition than that of NaNO_2_ [29]. More recently, the nitrite inhibitory effect has been studied based on an ion exchange principle; it was found that LDH nanocarriers can absorb Cl^−^ ions and release OH^−^ and NO_2_^−^ ions at the same time, increasing alkalinity and inhibition properties as a result [30].

Herein, we show the successful synthesis of nano-sized LDH loaded with a green corrosion inhibitor that can be released according to environmental conditions and migrate within the mortar to thoroughly coat the steel reinforcing bars. The anticorrosive properties were studied by using s sample of reinforced mortar as a representative structure in order to simplify the validation studies for the heterogenous concrete. To the best of our knowledge, the use of LDH loaded with migrating corrosion inhibitors to protect reinforced mortar has not yet been reported in the literature.

We will examine in detail the peculiar physical and chemical features of the glutamine-loaded LDH (Mg–Al–Gln) system and its ability to release a migrating corrosion inhibitor under corrosive conditions. The protection mechanism was investigated based on the variation of the corrosion resistance. Besides the absorbing chloride and the inhibitor anion releasing, LDH nanostructures can also act as a strong physical barrier, preventing the attack of aggressive species coming from the environment. Therefore, Mg–Al–Gln is proposed for a controlled release of the inhibitor able to protect steel reinforcements in a chloride-contaminated mortar structure in the long term.

## 2. Results

### 2.1. Corrosion Inhibitor Performance in Simulated Concrete Pore Solution

Preliminary information about the protective properties of the two selected corrosion inhibitors, pAB and Gln, was obtained by analyzing the changes that occur on the steel surface after their immersion in simulated concrete pore solutions. Different chemical environments were considered by comparing solutions without and with chloride ions at two different concentrations (0.01 or 0.15 wt.%, respectively).

The tests for the evaluation of the inhibitor performance were first carried out on simulated concrete pore solutions containing pAB or Gln without the presence of chloride ions. Moreover, a solution without the inhibitor was used as reference.

The surface of the steel substrates was characterized by optical microscopy after immersion in the simulated concrete pore solutions for 20 h, and both the inhibitors were still able to hinder corrosion processes. Some changes were observed only for the reference solution without corrosion inhibitor. By prolonging the tests up to 6 days, the degradation of the substrate in the absence of inhibitor becomes more pronounced. Under these conditions, all the inhibitors can protect the steel substrate as revealed by the optical images shown in Figure 1a.

When the tests were conducted in simulated concrete pore solutions containing a low amount of chloride ions (0.01 wt.%), both pAB and Gln inhibitors were still able to prevent the substrate modifications (Figure 1b).

To further investigate the behavior of the inhibitors in a corrosive environment, the treatments were also carried out in solutions containing a higher content of chloride ions, i.e., 0.15 wt.%, because by increasing the concentration of chloride, the depassivation of rebars is accelerated. Even under these aggressive conditions, the optical images of the steel substrates treated with Gln solution show that the inhibitor can hinder the onset of chloride-induced corrosion (Figure 1c). On the contrary, surface alterations occur during the same treatment in the pAB solution. The results converge to reveal the superior protective efficacy of Gln with respect to pAB in a more aggressive environment.

The tests continued for another 42 days to obtain further information on the inhibiting performance after a prolonged exposure to a degrading environment. At the end of the test, the steel surface was investigated via optical and SEM analysis (see Figure 2). As expected, the degradation of the substrate immersed in the pAB solution is more pronounced after a prolonged duration of the treatment with respect to the Gln inhibitor (Figure 2a,b) indeed. Remarkably, Gln is still able to protect steel and to hinder any corrosion processes. The absence of surface damage was confirmed via both optical and SEM micrographs (Figure 2c,d).

### 2.2. Corrosion Inhibitor Release Performance

The protective efficacy of the pAB and Gln inhibitors was also studied by carrying out using electrochemical measurements. The inhibition performance of both amino acids was evaluated according to the open circuit potential (OCP) and linear polarization resistance (LPR) measurements of the steel specimens in a synthetic pore solution [31,32,33].

The OCP and LPR measurements were carried out by using Gamry interface 1010 under room temperature (RT) and the local laboratory environment. A conventional three-electrode electrochemical cell was employed for measuring LPR with a steel-working electrode, a platinum counter electrode, and Hg/HgO as the reference electrode. Polarization sweeps from −20 to 20 mV relative to OCP were applied at a rate of 0.167 mV/s. Four groups of testing solutions were prepared: (i) Ca(OH)_2_ (as reference); (ii) Ca(OH)_2_ + 0.1 M inhibitors (as another reference); (iii) Ca(OH)_2_ solution + NaCl; and (iv) Ca(OH)_2_ solution + 0.1 M inhibitors + NaCl. 

It should be noted that the NaCl was added progressively as a source of Cl^−^ contamination in the test solution when the steel electrode reached the passivating state. The concentration of NaCl in the testing solutions was progressively increased in several steps every 24 h, from 0.1 M up to 0.3 M. The test cells used are 200 mL plastic bottles where 120 mL test solutions have been filled. The three electrodes were stabilized through the lid of the bottle, on which an additional hole was drilled for adding NaCl and to maintain the circulation of the air during the test. NaCl crystals were first dissolved with part of the test solution with an extra plastic cup. After this, it was transferred back to the bulk solution in the test cell and mixed thoroughly, blended with a plastic pipette. There was no stirring application during the whole test period. The OCP of the steel electrode in testing solution was recorded every 2 h, whereas LPR was recorded every 6 h up to 168 h.

The evolution of OCP was monitored to determine whether corrosion occurred. Consequently, in an alkaline solution containing chloride, corrosion (localized corrosion) was initiated and monitored after the disruption of the protective passive coating layer on the steel surface.

Figure 3a shows the evolution of the OCP of the steel in simulated concrete pore solution (blue dot), a mixed solution of simulated concrete pore solution and 0.1 M pAB (red dot), and a mixed solution of simulated concrete pore solution and 0.1 M Gln (black dot), containing chlorides with stepwise-increased concentrations ranging from 0.05 to 0.3 M. The steel in the pAB solution maintained its passivation status with the addition of chloride to a level of 0.1 M, as indicated by more positive OCP values than −270 mV. However, the addition of 0.3 M chloride caused a drop of the OCP to −287 mV at 108 h. This value dropped to −500 mV at 162 h at the end of measurements. 

These findings reveal that active corrosion has been developed and that the concentration at which the steel could sustain its passivity in pAB solution should range from 0.1 M to 0.2 M.

Figure 3b presents the evolution of the corrosion current density (defined as icorr) of the steel in the test solutions containing 0.1 M inhibitors with the addition of chloride. During the immersion of 48 h without any chloride addition, icorr, in all the solutions, retained a value raging from f 0.9 to 1.9 μA/cm^2^, clearly revealing the protective presence of a passive outermost layer.

The OCP remained positive above –270 mV in the NaOH solution containing Gln up to 144 h of immersion when the chloride concentration was 0.3 M. This result indicates that a stable passive layer has been formed on the steel surface, whose protective role is not affected by the presence of the chlorides. Simultaneously, the corrosion current (icorr) has been calculated by utilizing the polarization resistance (Rp) obtained from the elaboration on the LPR measurement. Andrade et al. [34] claim that these icorr values are passive in the reinforcement of steel because of the low loss of material produced.

After adding 0.1 M of chloride, the icorr increased markedly for pure Ca(OH)_2_ solution (~10 μA/cm^2^), which is indicative of a very active corrosion. Similarly, the icorr value for solutions containing pAB started to slightly increase from 0.40 (72 h) to 3.46 μA/cm^2^ (108 h) upon the addition of chloride at 0.2 M, resulting in a fluctuating increase in icorr as additional chloride was added. On the contrary, the icorr was stable by using Gln (black point), thus indicating the better protection property of this inhibitor with respect to the pAB material.

The above-presented OCP and LPR results have indeed demonstrated the best anti–corrosion performance of Gln inhibitor. Therefore, the Gln was selected as the most promising candidate to reliably protect the steel rebars in the synthesis of LDH nanocarriers.

### 2.3. Characterization of the Synthesized LDH

The X-ray diffraction patterns for Mg–Al–Gln and Mg–Al–NO_3_ are reported in Figure 4. Attention has been first focused on the XRD signal of the basal peak (003) located at about 2 theta 10°, which is influenced by the distance between the two adjacent metal hydroxide layers in the LDHs crystal lattice [35]. Mg–Al–NO_3_, was shown for comparison with Mg–Al–Gln and identified via the typical pattern of layered material (Figure 4a, red line) with a basal spacing in accordance with the presence of interlayer nitrate anions [36].

The X-ray diffraction patterns of LDH with nitrate anions are characterized by sharp diffraction peaks due to the high crystallinity of the products, unlike the glutamine derivative that produces a large basal reflection. The observed data showed a basal spacing of 8.8 Å for intercalated Mg–Al–NO_3_, decreasing to 7.8 Å for the LDH intercalated with glutamine anions (Figure 4a line black). Basal spacing following glutamine ion intercalation was found to be in the range of previously reported values [37,38].

Functional group information for Mg–Al–NO_3_ and Mg–Al–Gln was obtained via ATR–FTIR spectroscopy, and their spectra are shown in Figure 4b. A broad band between 3500 and 3200 cm^−1^ is observed, representing the stretching vibrations of the hydrogen-bound hydroxyl group of the two layers of hydroxide and the water between the layers. The strong absorption band at 1343 cm^−1^ is related to the vibration mode of the interlayer nitrate anion. The bands between 1760–1500 cm^−1^ and 1200–1000 cm^−1^ (green box) reveal the clear presence of Gln, which is present as an interlayer anion, thus demonstrating that the loading process has occurred [39].

By comparing the FT–IR spectra of the Mg–Al–Gln materials with those of the nitrate hydrotalcites, it may be concluded that there has not been a complete loss of intercalary nitrate anions and bound water. This is revealed by the presence of the *ν*_3_ stretching mode band of NO_3_^−^ located at around 1350 cm^−1^ and the O–H stretching band at 1650 cm^−1^ for interlayer water molecules. This observation is in good agreement with the XRD information.

Figure 5 shows a representative SEM micrograph of the LDH particles deposited on a single crystal silicon substrate from the LDH suspension. The LDHs appear to be composed of round-edged, aggregated, non-uniform particles, which is probably due to the presence of organic molecules in interlamellar space, which may affect the superficial interaction between particles. The dimension of platelets ranges from 50 nm to 200 nm [40], thus confirming that a nanosized structure was obtained.

TG and DSC were employed to investigate the thermal behavior of glutamine and modified hydrotalcites—particularly, to quantify the loading efficiency of the Mg–Al–Gln system. The pure glutamine thermal behavior has been also studied as a comparison. The measured TG/DSC curves are shown in Figure 6. The experiments have been carried out under a nitrogen atmosphere in a temperature range of 25 to 1000 °C. The thermograms of pure glutamine reveal that the most significant mass loss ranges from 180 to 481 °C, with a sharp endothermic signal peak at about 200 °C. These values are comparable with those previously reported in the literature [41,42].

The thermogram of the Mg–Al–Gln system shows, in the range of 40 to 170 °C, weight loss of about 10%, which corresponds to the elimination of the water both physiosorbed and intercalated. The second and third weight loss signals in the 176–246 °C and 246–481 °C ranges, respectively, can be attributed to a concomitant decomposition of intercalated guest anions and a dihydroxylation phenomenon. Furthermore, as can be seen from the associated DSC thermograms, the nature of the interlayer guest anion a markedly influences the thermal behavior in the second stage of the decomposition process.

### 2.4. Inhibitor Release and Protective Properties of LDH

A metal corrosion processes generally occurs in the presence of aggressive anions, such as Cl^−^ or SO_4_^2−^, to hinder the degrading phenomena; hence, we require suitable and reliable inhibitors able to reduce, slow down, or prevent the negative role of these anions. To confirm the positive stimuli responsive properties of the Mg–Al–Gln system, we have prepared three different solutions containing 0.35 g of Mg–Al–Gln in 10 mL of solution: (1) water or solutions; (2) alkaline solution; and (3) alkaline solution with chloride ions (0.3 M NaCl).

After the release of the corrosion inhibitor (occurred in 120 h), the three supernatants were added to 4.5 mL of ninhydrin to promote the formation on Ruhemann’s purple dye. Thanks to the previously built calibration line, it was possible to trace the signal intensity back to the amount of inhibitor released in each solution and calculate the percentage of inhibitor released (Figure 7a). The release curves of glutamine from Mg–Al-layered double hydroxide in NaCl solutions were determined and are shown in Figure 7a.

The inhibitor release was rapid in the first hours of immersion (3 h) and then slowed down. As expected from the proposed anion exchange mechanism, the presence of chlorides led to the release of amino acids. The glutamine release behavior was studied in the neutral solution and the simulated concrete pore solution for comparison.

The results clearly show that the concentration of glutamine does not increase with time via treatments at neutral or alkaline pH when chloride ions are not present.

It can be observed that only a low amount of glutamine is released after 120 h of treatments in the solutions without chloride; in the presence of chloride, the glutamine is released in greater amounts after treatments for 120 h until it stabilizes (amount to 14% of the total content). These results confirm that the release of glutamine is based exclusively on an exchange reaction between amino acid and chloride ions, but they do not explain what happens to the remaining 14% (on a total of 28% in loading).

In order to gain further insights into the release mechanism, the Mg–Al–LDH sample treated with 3.5% NaCl solution was recovered and analyzed via DTA (Figure 8a), FTIR (Figure 8b), and XRD (Figure 8c).

As shown in Figure 8b, the characteristic peaks of the glutamine anion at about 1760–1500 cm^−1^ can be observed also after the release process (green box), indicating that a certain amount of this anion is still within the LDH nanocontainer. These results show that the release is promoted by the presence of chloride, even if the information is only qualitative.

In order to compare the agreement between the amount of inhibitor loaded in the nanocarrier at the start of the study (around to 28% in loading) and the amount of inhibitor released during the experiments with simulant solutions and Chloride (around to 14%), DTA analysis has been carried out on Mg–Al–Gln after the release test (Figure 8a).

The percentage calculated for the precipitate is equal to 14.53% at loading. The consistency of the percentage values calculated via DTA and UV–Vis analyses is good. However, when the percentages related to the amount of inhibitor released in the Ca(OH)_2_ + NaCl solution (14% in loading) and the percentage related to the amount of inhibitor still loaded on the nanocarrier after the releasing tests (14.53%) were added together, a value other than that predicted by the initial DTA measurements of the inhibitor initially loaded on the nanocarrier (around to 28%) was obtained.

The distinctive peaks in Mg–Al–Gln XRD patterns shifted after chloride exchange, as seen in Figure 8c. The diffraction peaks agreed well with those characteristics of LDH-type materials reported in the literature [43]. The interlayer distance of Mg–Al–Cl can be determined as 7.81 Å based on the (003) diffraction peak.

Corrosion experiments were conducted on reinforced mortar specimens in order to evaluate the effectiveness of Mg–Al–Gln as a smart inhibitor. They were designed using a 6 mm diameter low carbon steel rebar as the working electrode and a graphite rebar as the counter electrode. An external Hg/HgO was used as the reference electrode (RE).

Electrochemical impedance spectroscopy (EIS) was performed using a three-electrode arrangement and a frequency response analyzer. The impedance response was recorded over 75 days at time intervals of 0 h, 2 h, 5 h, 24 h, 72 h, 144 h, 288 h, 624 h, 912 h, 1272 h, and 1800 h in saturated saline solution with 0.3 M NaCl.

In Figure 9, the OCP evolution of steel bars embedded in mortar specimens and subjected to chlorides penetration is reported. Each value has been averaged from the results of three specimens.

The results reveal that the open circuit potential of the rebar RIF (red point) decreases fast from −20 to −430 mV within 24 h of the corrosion. Mg–Al–Gln showed more positive potential values in the range of around −168 mV after 150 h, suggesting its anti–corrosion performance. The glutamine anion released can significantly reduce carbon steel corrosion.

Figure 9b shows the impedance response (plotted in the Nyquist diagram) for all specimens throughout a range of time measurements and displays varying trends over time. The total impedance of the RIF system begins to fall after 24 h of immersion, revealing an increase in increase in corrosion activity on the steel surface. In contrast, the EIS spectrum of Mg–Al–Gln shows no apparent change within the first 144 h. The creation of a passive layer on the steel surface as a result of the inhibitors’ release causes an increase in impedance. These changes clearly show that the inhibitor diffuses through the mortar, inhibiting corrosion activity on the steel surface. However, after 1000 h of treatments, the passive film became unstable due to the presence of chloride ions, and the corrosion process started, leading to a decrease in the overall impedance values.

Representative equivalent circuits were utilized to fit the EIS data (Figure 9b inset) to obtain quantitative parameters from the EIS diagrams [44,45,46]. The electrolyte resistance R0 is connected in series with other elements in the model. R1 resistance is related to the resistance of products generated in cement pores around steel reinforcement, and Rt resistance is related to the corrosion reaction, which corresponds to Rp in this case.

The values of the electrical parameters obtained by fitting the EIS data are shown in Table 1. The EIS diagrams displayed a good fit with the suggested equivalent circuits, with chi–squared (χ^2^) values in the order of 10^−3^ and 10^−4^ expressing the goodness of fit. By fitting the impedance spectra to the equivalent circuit presented in inset Figure 9, the electrochemical parameters for NO INH and Mg–Al–Gln at 288 h were determined. Table 1 summarizes the obtained values. Rt data differences for the same time immersion can be attributable solely to steel surface corrosion. 

The Rt values obtained via the fitting process appear useful for characterizing the degree of corrosion of rebar. Table in Section 3 shows that the Rt values of the NO INH sample and the Mg–Al–GLN sample decreased with increasing values of the w/c ratio. Assuming that R*_ct_* is inversely proportional to icorr, the R*_ct_* values would indicate, in agreement with the results of DC measurements, that icorr increases with the w/c ratio. Icorr values estimated from the R*_ct_* values using a Stern–Geary constant of 27 mV were in reasonably good agreement with those in Figure 3b.

Thus, the available results give a qualitative indication that the LDH nanocarriers are useful for inhibiting diffusion into a mortar matrix. To sum up, the results of Ecorr monitoring, LPR, and EIS show good agreement and suggest that these techniques can be used together to monitor the corrosion development of reinforced specimens exposed directly to simulate corrosion solutions.

## 3. Experimental Section

The functionalization of layered double hydroxide (LDH) with glutamine (Gln) was performed by an anion exchange method from the corresponding LDH intercalated with nitrate. The latter was synthesized by using the co-precipitation method. The intercalation of anionic molecules into LDH by anion exchange is the most common procedure for their functionalization [47], as shown in our previous work [17].

In a typical synthesis, the mixed nitrate solution containing Mg(NO_3_)_2_ and Al(NO_3_)_3_·9H_2_O was added dropwise to the amino acid solution at room temperature with stirring; then, the resulting precipitate was washed extensively with water, centrifuged, and dried at 70 °C under vacuum for a further 18 h. The so-obtained Mg–Al–NO_3_ was then dried at 100 °C in an oven for 12 h and used for comparison during the characterizations.

The optical microscopy (OM) characterization was carried out to observe modifications occurring at the surface of the steel after accelerated corrosion tests in order to explore the protective performance of the two selected inhibitors. The steel disks were analyzed via the mean of optical microscopy after 40 h, 6 days, 20 days, and 5 months of immersion [17]. OM investigations were performed using a Leica MZFLIII microscope equipped with a digital camera (Leica DFC 320).

Amino acid inhibition performance was assessed based on open circuit potential (OCP) and linear polarization resistance (LPR) measurements of the steel specimens in a simulated concrete pore solution obtained from a saturated Ca(OH)_2_ aqueous solution containing NaOH 0.01 mol/L (pH 12.6), which has been used in earlier research to simulate the alkaline concrete pore solution [6,30,31].

The OCP and LPR measurements were performed using Gamry Interface 1010 at room temperature (RT) and in a local laboratory environment. A conventional three-electrode electrochemical cell was employed for measuring LPR with a steel-working electrode, a platinum counter electrode, and Hg/HgO as the reference electrode. Polarization sweeps ±20 mV relative to OCP was applied at a rate of 0.167 mV/s. The chemical composition of the steel B450C used in the investigation was (in mass%): 0.16C, 1.02Mn, 023Si, 0.019P, 0.041S, 0.52Cu, 0.20Cr, 0.23Ni, 0.06Mo, 0.005V, and 0.0097N.

Corrosion tests were also performed via the immersion of steel disks in a simulated concrete pore solution. Chloride ions were added to the solution to promote the corrosion, and the modifications occurring at the metal surface were periodically observed via optical microscopy. The concentration of the inhibitor was 50 mM for both pAB and Gln solutions. Regarding the simulated concrete pore solution, this was prepared by dissolving 0.001 wt.% Ca(OH)_2_ in deionized water and fixing the pH at 10.5 using KOH [17]. The immersion tests were performed at room temperature (r.t.).

The morphological properties of LDH materials were investigated by means of high-brilliance and high-spatial-resolution LEO Gemini 1530 (Zeiss, Jena, Germany) field emission scanning electron (FE-SEM) equipped with an INCA 450 energy dispersive X-ray spectrometer (EDS) and a four-sector backscattered electron detector (BSD).

The chemical functional groups of synthesized Mg–Al–Gln were analyzed via Fourier Transform Infrared spectroscopy. The spectra were documented using a Nicolet iS50 spectrometer (Thermo Fisher, Rodano, Italy) equipped with an ATR accessory, and the measurements were recorded with a diamond crystal at 32 scans with resolution of 4 cm^−1^.

X-ray diffraction (XRD) and thermal analysis were used to study the chemical composition of Mg–Al–Gln. XRD measurements were performed in a Siemens 5000 X-ray powder diffractometer with a Cu Kα radiation (λ = 1.5418 Å) filtered by a nickel window. Angular values with a step size of 0.05° and a sampling time of 2 s were the experimental parameters used for data acquisition. 

Mg–Al–Gln was characterized via thermogravimetric analyses, TGA, to assess the amino acid content of the interlayer. TGA characterization was recorded with an automated thermal analyzer Model SDT Q600 (TA Instruments, New Castle, DE, USA) in air atmosphere (gas fluxes of 100 mL/min).

The stimuli response properties of Mg–Al–Gln-based materials were investigated by monitoring the release of inhibitors in solutions at neutral pH and at alkaline pH in and without the presence of chloride ions. Because of the amino acid’s inability to be detected via UV–Vis spectroscopy, it was necessary to carry out preliminary experiments.

Hence, the release of glutamine was monitored using the ninhydrin test, which consists of the formation of a colored product called Ruhemann’s purple (RP), caused by the reaction of the inhibitor with a certain amount of ninhydrin added to the dispersion [35]. This is a general test that can be used to detect all the amino acids.

Measuring the color yield at several reaction times shows the release rate and whether the maximum release has been reached. A calibration curve has been obtained, acquiring different spectra from water solutions (pH = 6) at different glutamine concentrations. Figure 10 shows the spectra obtained from UV–Vis at different glutamine concentration solutions used for the building of the calibration line (b).

Non-Standard mortar specimens (cubes of 10 × 10 × 10 cm^3^) with a sand/cement ratio of 5 and a water/cement ratio of 0.7 were produced. The cement used to prepare the mortars was a CEM I 42.5R (supplied by Cementos Portland Valderrivas, Pamplona, Spain), and the sand was normalized sand DIN EN 196–1 (particle size between 0.08 and 2 mm). Mortars were prepared with one rebar of low carbon steel (as a working electrode) with a diameter of 6 mm and a second rebar of graphite as a counter electrode. Reinforced mortars were cured for 28 days in the climatic chamber (21 °C, 99% RH).

Additionally, non-standard mortar prismatic specimens (4 × 4 × 16 cm^3^) with the same dosage as the previous specimens (sand/cement ratio 5, w/c 0.7) were prepared and cured at the same conditions (28 days in the Climatic Chamber at 21 °C, 99% RH). The mortars were then mechanically tested using an Ibertest (Autotest–200/10–SW) test frame (3 replicates per set). The microstructure of the mortars was investigated via mercury intrusion porosimetry (MIP) on a Micrometrics AutoPore IV 9500 V1.09 pore sizer and by soaking the mortars in isopraponal for two days and then drying them in a desiccator prior to analysis. The results of this characterization are summarized in Table 2.

## 4. Conclusions

Green inorganic nanocarriers, such as LDH, and corrosion inhibitors, such as pAB and Gln, have been selected for use in preparing smart metal corrosion-inhibiting material to hinder the steel degradation phenomena at the high pH values of those mortars. 

The efficiency of loading corrosion inhibitor compounds via inorganic nanocarriers has been investigated using FTIR spectroscopy, while thermogravimetric analysis has been used to quantitatively evaluate the amount of corrosion inhibitor loaded by the smart systems. Furthermore, the ninhydrin test was implemented to estimate the amount of released inhibitor. 

The findings have demonstrated that that the anticorrosive capacities of Mg–Al–Gln can be achieved by applying the nanocarrier on the surface of reinforced mortars. In this way, the inhibitor released from Mg–Al–Gln can migrate and successfully inhibit the steel degradation phenomena. 

The proposed approach is very promising for the non-invasive corrosion protection of reinforced structures, and anticorrosion improvement can be achieved without damage to the concrete matrix. This significantly simplifies the entire trial. This improvement was attributed to the chemical absorption of the migrating inhibitor on the embedded steel rebars and possibly to a sealing impact on the mortar pores (i.e., the pore-blocking effect).

Because of the small number of smart nanocarrier systems applied to the surface of the specimens, the mortar is almost unaffected, and this is a very attractive option for the corrosion protection of already existing structures due to its cost-effectiveness and ease of application.

## Figures and Tables

**Figure 1 molecules-28-05863-f001:**
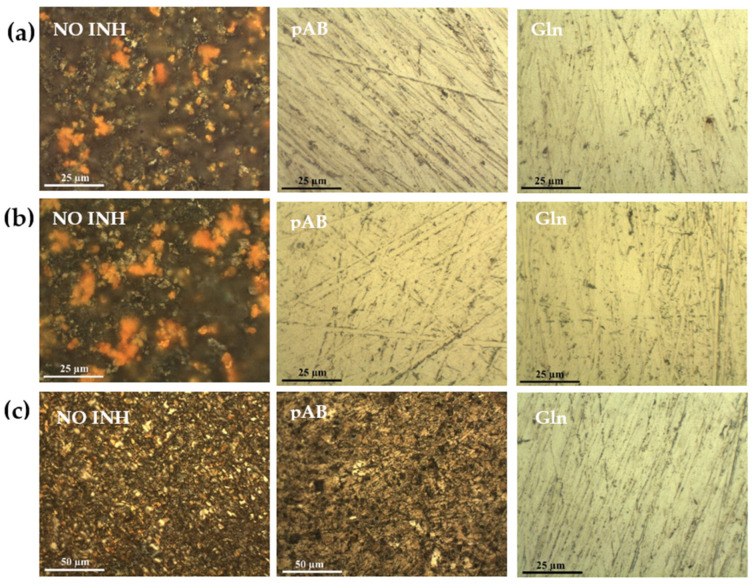
Optical images of the steel surface treated for 6 days at r.t. in the simulated concrete pore solutions without inhibitor (NO INH), with pAB 50 mM, and with Gln 50 mM. The tests were conducted in solutions: (**a**) without chloride ions; (**b**) with 0.01 wt.% of chloride ions; and (**c**) with 0.15 wt.% of chloride ions.

**Figure 2 molecules-28-05863-f002:**
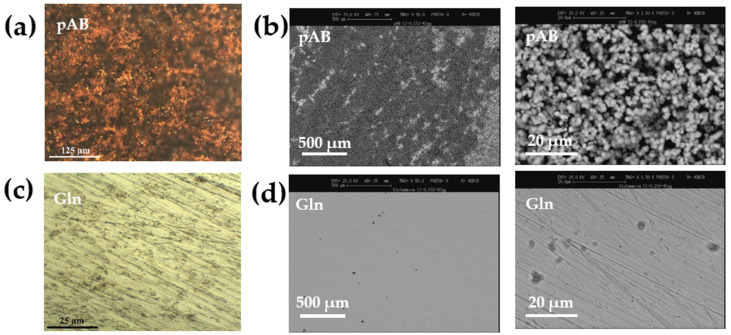
Optical and SEM images of the steel surface treated for 42 days at r.t. in the simulated concrete pore solutions containing 0.15 wt.% of chloride ions and the corrosion inhibitor, namely, (**a**) optical and (**b**) SEM images after treatment in solutions containing pAB 50 mM; and (**c**) optical and (**d**) SEM images after treatment in solutions containing Gln 50 mm.

**Figure 3 molecules-28-05863-f003:**
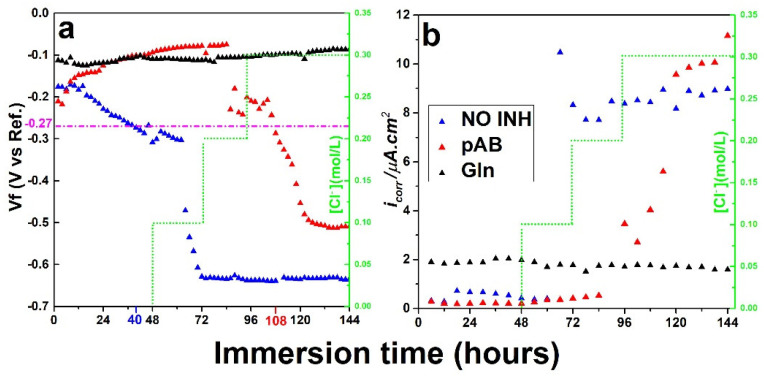
(**a**) Evolution of OPC of the steel (OCP) in simulated concrete pore solution without inhibitor (NO INH), with 0.1 M pAB (pAB). and with 0.1 M Gln (Gln). Chloride concentration was stepwise-increased in a range from 0.05 up to 0.3 M. The results above the pink line indicates that a stable passive layer has been formed on the steel surface. (**b**) Corrosion current density (icorr) evolution of steel in a simulated concrete pore solution without inhibitor (NO INH), with 0.1 M pAB (pAB), and with 0.1 M Gln (Gln).

**Figure 4 molecules-28-05863-f004:**
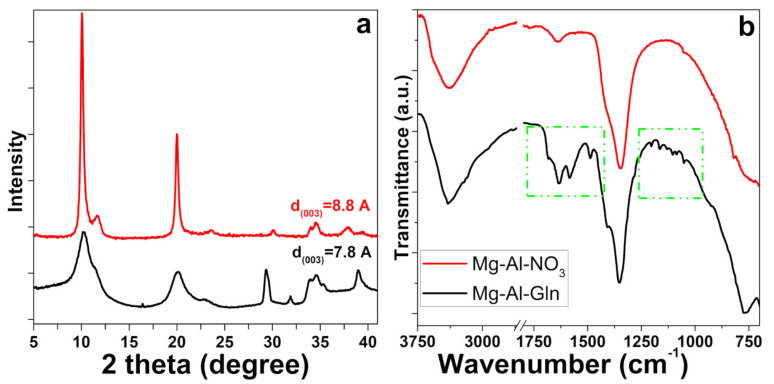
(**a**) X-ray diffraction pattern of Mg–Al–Gln (black line) and Mg–Al–NO_3_ (red line); (**b**) ATR–FTIR spectra of Mg–Al–NO_3_ (red line) and Mg–Al–Gln and (black line).

**Figure 5 molecules-28-05863-f005:**
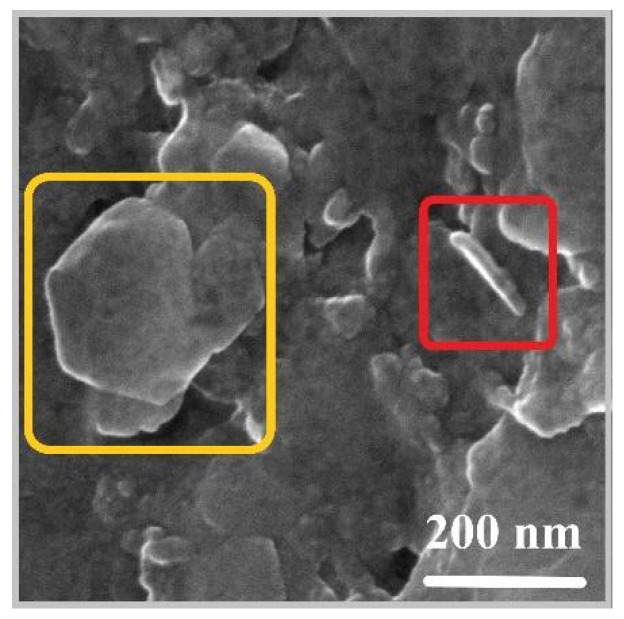
FEG–SEM images of Mg–Al–Gln. Hexanol-shaped platelets are visible, with lateral dimensions ranging over a few hundred nanometers (yellow box) and thicknesses of several nanometers (red box).

**Figure 6 molecules-28-05863-f006:**
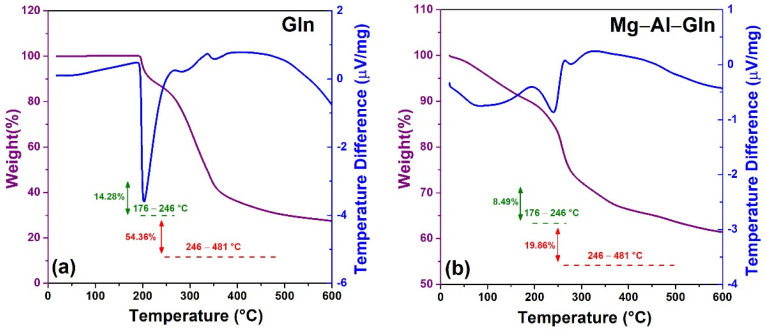
Thermogravimetric (TG) curves in the N_2_ flow of pure glutamine (**a**) and Mg–Al–Gln (**b**).

**Figure 7 molecules-28-05863-f007:**
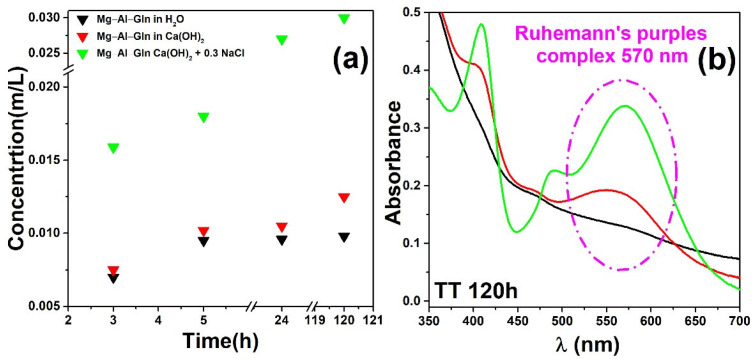
(**a**) Concentration of glutamine released in aqueous solutions at neutral pH in an alkaline solution and in an alkaline solution containing chloride ions. (**b**) UV–Vis spectra obtained from three different solution of Mg–Al–Gln and ninhydrin in water (black line), in Ca(OH)_2_ (red line), and in Ca(OH)_2_ + NaCl (green line), prepared as described above three hours after their preparations.

**Figure 8 molecules-28-05863-f008:**
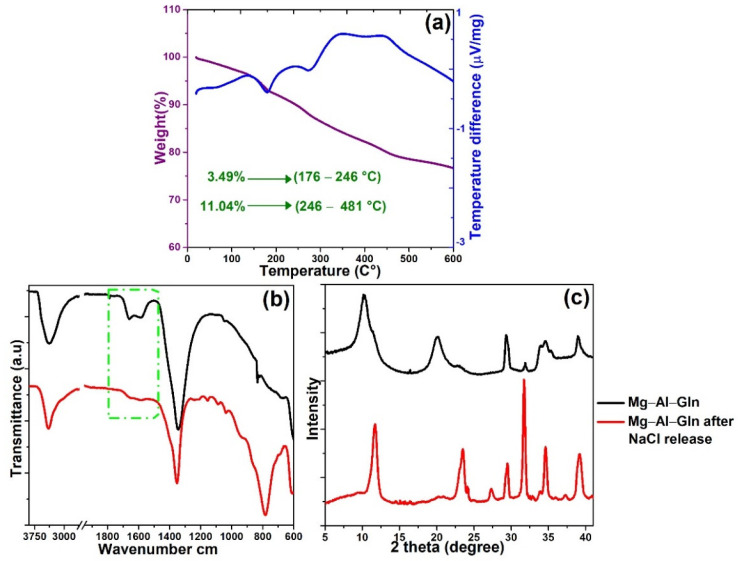
(**a**) DTA spectrum of the precipitate sampled from the Ca(OH)_2_ + NaCl solution with dispersed Mg–Al–Gln; (**b**) ATR–FTIR spectra of Mg–Al–Gln and sample reclaimed from NaCl solutions contain Mg–Al–Gln; (**c**) XRD pattern of Mg–Al–Gln and sample reclaimed from NaCl solutions contain Mg–Al–Gln.

**Figure 9 molecules-28-05863-f009:**
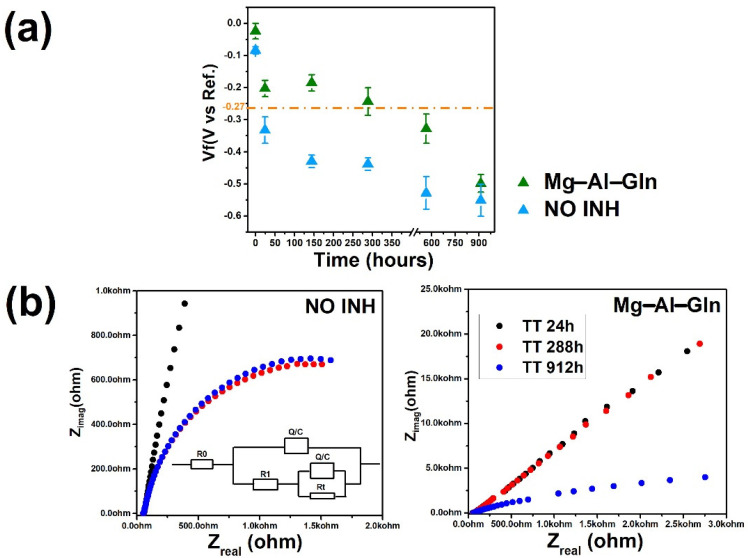
(**a**) The OCP/Ecorr (average of three specimens) evolution obtained using Hg/HgO as the reference electrode for rebars embedded in mortar specimens in the chloride migration test. (**b**) Nyquist spectra (**a**,**b**) for RIF and Mg–Al–Gln after 24 h, 144 h, 288 h, and 912 h of immersion in NaCl solution (0.3 M). Schematic representation of the circuit during IR compensation (inset).

**Figure 10 molecules-28-05863-f010:**
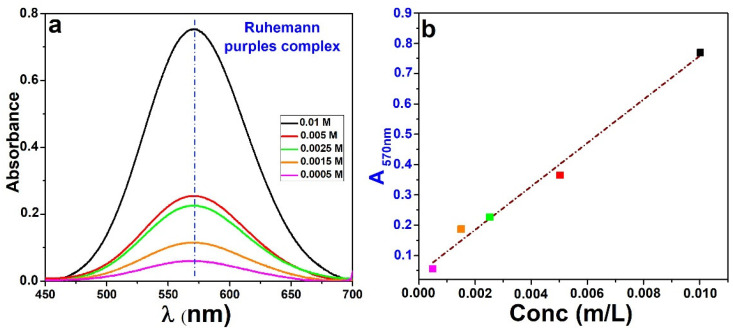
(**a**) UV–VIS spectra obtained from different glutamine concentration solutions. (**b**) Calibration line.

**Table 1 molecules-28-05863-t001:** The fitting results of EIS spectra of the carbon steel in RIF and Mg–Al–Gln.

SAMPLES	h	R0	C	R1	W/O	CPE	n	Rt	Χ^2^
RIF.	288	77.5	2.43 × 10^−5^	2.73	9.5143 × 10^−3^	8.5143 × 10^−4^	0.95	4.2143 × 10^4^	3.0443 × 10^−5^
Mg–Al–Gln	288	68.4	6.5543 × 10^−5^	2.896	1.5143 × 10^−2^	1.8643 × 10^−3^	0.81	1501	2.0343 × 10^−4^

**Table 2 molecules-28-05863-t002:** Mechanical strengths and porosity analysis of the non-standard mortars.

Non Standard Mortar	Comprehensive Strengths (MPa)	Total Porosity (%)	Pore Size Distribution (%)					
	26.9 ± 2.03	14.90	>100 µm	100–10 µm	10–1	1–0.1	0.1–0.01	<0.01
			0.55	0.27	0.82	7.78	5.12	0.34

## Data Availability

Any data or material that support the findings of this study can be made available by the corresponding author upon request.

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
