# Peer review of "Smart Inhibition Action of Amino Acid-Modified Layered Double Hydroxide and Its Application on Carbon Steel"

_molecules, 2023, doi:10.3390/molecules28155863_

Round 1

Reviewer 1 Report

The manuscript by Messina et al describes the use of Gln as a non-toxic corrosion inhibitor for steel at alkaline pH. Since most studies in this field focus on media of acidic pH, the study adds some important findings to the literature. However, the manuscript has not been assembled very carefully and needs to be revised significantly before publication. I have not corrected any typos or grammatical errors. However, there are numerous mistakes in the manuscript and I recommend proofreading by a native speaker. Several passages are hard to understand. In addition:

-the term “smart inhibition action” is somewhat exaggerating the findings presented, I recommend to rephrase the title

-explain all abbreviations in abstract: eg RP

-use correct three letter code: Gln not GLN

-give details about the steel used in this study, type of steel, supplier etc. More generally: I recommend to give more details for almost any of the experiments performed.

-explain all abbreviations in the text: eg pAB (p-aminobenzoic acid?)

-add scale bars to microscopy images (Fig. 2), same for Fig. 3 a and c

-Caption of Fig. 2: give more information! What concentration of the inhibitor was used? At what temperatures have the experiments been done?…

-lines 213-214: “…all the inhibitors were still able to hinder corrosion processes…” this suggests to the reader, that several inhibitors were tested. However, as far as I understand only two inhibitors are investigated here (Gln and pAB). “Both” instead of “all” might be a better wording.

-the long term corrosion experiments should be analyzed by gravimetry to support and quantify the anticorrosive effects

-studies of amino alcohols and amino acids as corrosion inhibitors at alkaline or neutral pH are comparatively rare, the authors might therefore consider the following recent additional references in this context: DOI: 10.1007/s00726-023-03260-x, DOI: 10.1007/s00726-023-03260-x, DOI: ARTN 110.3390/computation9010001, DOI: 10.1002/maco.202213406, Mater Sci Eng 8(1): 1-14.

I have not corrected any typos or grammatical errors. However, there are numerous mistakes in the manuscript and I recommend proofreading by a native speaker. Several passages are hard to understand.

Author Response

Dear Editor,

Please find attached the point-by-point response to the reviewers and the list of amendments made to the manuscript.

Sincerely

The authors

Reviewer #1:

Comments and Suggestions for Authors

The manuscript by Messina et al describes the use of Gln as a non-toxic corrosion inhibitor for steel at alkaline pH. Since most studies in this field focus on media of acidic pH, the study adds some important findings to the literature. However, the manuscript has not been assembled very carefully and needs to be revised significantly before publication. I have not corrected any typos or grammatical errors. However, there are numerous mistakes in the manuscript and I recommend proofreading by a native speaker. Several passages are hard to understand.

The authors thank the Reviewer for the constructive inputs. Point by point responses are reported below.

In addition:

-Reviewer’s remark N.1:

The term “smart inhibition action” is somewhat exaggerating the findings presented, I recommend to rephrase the title

Author response and Amendments to the manuscript:

The results reported in 3.4 paragraph “Inhibitor release and protective properties of LDH” clearly show that the glutamine is not released by treatments at neutral or alkaline pH when chloride ions are not present. The protective effect of LDH is related to the stimulus-responsive properties of LDH that releases the Gln corrosion inhibitor only in the presence of chloride ions. Moreover, the entrapment of the aggressive chloride species into LDH structure decreases the corrosiveness of the medium. This double protective function, associated with the active mechanism of inhibition provided by the released amino acids to the exposed steel surface, gives a smart property to the LDH-Gln loaded, making it an active anticorrosion nanomaterial with great potential to be applied in corrosion protection.

-Reviewer’s remark N.2:

Explain all abbreviations in abstract: eg RP

Author response and Amendments to the manuscript:

Done

-Reviewer’s remark N.3:

Use correct three letter code: Gln not GLN

Author response and Amendments to the manuscript:

Done

Reviewer’s remark N.4:

Give details about the steel used in this study, type of steel, supplier etc. More generally: I recommend to give more details for almost any of the experiments performed

Author response and Amendments to the manuscript:

Done

Reviewer’s remark N.5:

Explain all abbreviations in the text: eg pAB (p-aminobenzoic acid?)

Author response and Amendments to the manuscript:

Done

Reviewer’s remark N.6:

Add scale bars to microscopy images (Fig. 2), same for Fig. 3 a and c

Author response and Amendments to the manuscript:

Done

Reviewer’s remark N.7:

Caption of Fig. 2: give more information! What concentration of the inhibitor was used? At what temperatures have the experiments been done?

Author response and Amendments to the manuscript:

Done

Reviewer’s remark N.8:

lines 213-214: “…all the inhibitors were still able to hinder corrosion processes…” this suggests to the reader, that several inhibitors were tested. However, as far as I understand only two inhibitors are investigated here (Gln and pAB). “Both” instead of “all” might be a better wording.

Author response and Amendments to the manuscript:

Done

Reviewer’s remark N.9:

The long term corrosion experiments should be analyzed by gravimetry to support and quantify the anticorrosive effects

Author response and Amendments to the manuscript:

The capacity of the developed systems to inhibit corrosion on steel is related both to the anticorrosion properties of the inhibitor and to the stimuli responsive properties of the nanocarriers. In this study, we first evaluate the inhibitors efficiency in simulated concrete pore solutions and then we demonstrate the stimuli responsive properties of inhibitor-loaded layered double hydroxides, thus highlighting their potentiality.

In addition, electrochemical methods were used to evaluate the protective performances, since they are typically considered to be efficient characterization and monitoring tools.

We thank the Reviewer for the suggestion and gravimetry measurements will be certainly used in a next project.

Reviewer’s remark N.10:

Studies of amino alcohols and amino acids as corrosion inhibitors at alkaline or neutral pH are comparatively rare, the authors might therefore consider the following recent additional references in this context: DOI: 10.1007/s00726-023-03260-x, DOI: 10.1007/s00726-023-03260-x, DOI: ARTN 110.3390/computation9010001, DOI: 10.1002/maco.202213406, Mater Sci Eng 8(1): 1-14.

Author response and Amendments to the manuscript:

Done

Reviewer 2 Report

Here authors report the synthesis of a magnesium-aluminum Layered Double Hydroxides, which can be used as migrating inhibitor on repair mortars specimens, storing amino acid as a green inhibitor Glutamine Layered Double Hydroxide (Mg-Al-GLN). Electrochemical techniques, measurements of corrosion potential and electrochemical impedance spectroscopy were used to determine the corrosion behaviour of the specimens when a cell containing a 3.5% NaCl solutions was applied in the rehabilitation mortar. Mg-Al-GLN presented the best corrosion protection performance, exhibiting an increase of RP for all the measured periods, compared to the reference (NO INH). This effect is a consequence of a double mechanism of protection provided by stimuli-response release of glutamine and removal of corrosive chloride species from the medium. So, the research work well done and discussed. It can be published in Molecules.

Author Response

Dear Editor,

Please find attached the point-by-point response to the reviewers and the list of amendments made to the manuscript.

Sincerely

The authors

Reviewer #2:

Here authors report the synthesis of a magnesium-aluminum Layered Double Hydroxides, which can be used as migrating inhibitor on repair mortars specimens, storing amino acid as a green inhibitor Glutamine Layered Double Hydroxide (Mg-Al-GLN). Electrochemical techniques, measurements of corrosion potential and electrochemical impedance spectroscopy were used to determine the corrosion behaviour of the specimens when a cell containing a 3.5% NaCl solutions was applied in the rehabilitation mortar. Mg-Al-GLN presented the best corrosion protection performance, exhibiting an increase of RP for all the measured periods, compared to the reference (NO INH). This effect is a consequence of a double mechanism of protection provided by stimuli-response release of glutamine and removal of corrosive chloride species from the medium. So, the research work well done and discussed. It can be published in Molecules.

The authors thank the Reviewer for the positive comments.

Round 2

Reviewer 1 Report

necessary corrections have been performed